# The Reversible Methylation of m6A Is Involved in Plant Virus Infection

**DOI:** 10.3390/biology11020271

**Published:** 2022-02-09

**Authors:** Jianying Yue, Yao Wei, Mingmin Zhao

**Affiliations:** College of Horticulture and Plant Protection, Inner Mongolia Agricultural University, Hohhot 010018, China; yuejianying2018@163.com (J.Y.); weiyao2018@163.com (Y.W.)

**Keywords:** m6A methylation, plant viruses, defense mechanism

## Abstract

**Simple Summary:**

N6-methyladenosine (m6A) is the most prevalent modification in the mRNAs of many eukaryotic species. The abundance and effects of m6A are determined by dynamic interactions between its methyltransferases (“writers”), demethylases (“erasers”), and binding proteins (“readers”). It has been indicated that there is a strong correlation between m6A and virus infection in mammals. In the case of plant virus infection, it appears that m6A plays a dual role. On the one hand, m6A acts as a plant immune response induced by virus infection, inhibiting viral replication or translation through methylation of viral genome RNAs. On the other hand, m6A acts as part of an infection strategy employed by plant viruses to overcome the host immune system by interacting with m6A-related proteins. We proposed that antagonists of m6A-related proteins might be used to design new strategies for plant virus control in the future.

**Abstract:**

In recent years, m6A RNA methylation has attracted broad interest and is becoming a hot research topic. It has been demonstrated that there is a strong association between m6A and viral infection in the human system. The life cycles of plant RNA viruses are often coordinated with the mechanisms of their RNA modification. Here, we reviewed recent advances in m6A methylation in plant viruses. It appears that m6A methylation plays a dual role during viral infection in plants. On the one hand, m6A methylation acts as an antiviral immune response induced by virus infection, which inhibits viral replication or translation through the methylation of viral genome RNAs. On the other hand, plant viruses could disrupt the m6A methylation through interacting with the key proteins of the m6A pathway to avoid modification. Those plant viruses containing ALKB domain are discussed as well. Based on this mechanism, we propose that new strategies for plant virus control could be designed with competitive antagonists of m6A-associated proteins.

## 1. The Discovery of N6-Methyladenosine (m6A) in Viral RNAs and in Plants

In the 1970s, m6A methylation was detected in viral systems when scientists were trying to characterize RNA modifications, including mRNAs and cap structures. The first description of m6A methylation was by Desrosiers in 1974, who discovered the methylated nucleosides in messenger RNAs from Novikoff hepatoma cells [1]. Not only in cellular mRNAs, m6A modification were also identified in viral RNAs such as influenza virus [2], HSV1 type [3], Rolls sarcoma virus [4,5], Simian virus 40 [6,7,8], and Adenovirus type 2 [9]. At that time, however, investigations were limited to descriptive reports due to the limitations of research technology such as chromatographic analysis of radiolabeled and enzymatically digested RNAs. For example, in Rolls sarcoma virus, there are seven m6A sites in the 1865 nt region of viral genomic RNA. The m6A methylation is located in the internal position of HSV mRNA. In simian virus, m6A residue is present in the coding region of the SV40 mRNA. The highest level of internal 6-methyl adenosine residue in simian virus 40 RNA is located on the L chain and E-strand RNA transcripts. In addition, the m6A-methylation affects the splicing of the Rous sarcoma virus Env mRNA [8]. All these viruses have a nuclear stage in their life cycle. This led virologists to believe that the nucleus was the primary site of m6A modification of RNAs.

In the following decades, research on m6A methylation in viruses has slowly progressed. In 2012, genome-wide distribution of m6A was revealed with the development of m6A-specific binding immunoprecipitation next-generation sequencing method (MeRIP-SEQ). Meyer and others comprehensively analyzed the distribution of m6A methylation in a human genome [10]. At the same time, Dominissini et al., determined the distribution of m6A methylation in the entire transcriptome of mice, proving that the distribution of m6A in humans and mice is highly conserved [11]. This greatly promoted the understanding of m6A methylation in viruses as well. In recent years, with the development of multiple detection technologies, studies on m6A methylation achieved rapid developments. The abundance and effects of m6A on RNAs are determined by the dynamic interplay between its methyltransferases (“writers”), binding proteins (“readers”), and demethylases (“erasers”) [12,13]. Apart from messenger RNAs (mRNAs), transfer RNAs (tRNAs), ribosomal RNAs (rRNAs), small non-coding RNAs, and long non-coding RNAs (lncRNAs) can be modified by m6A [14,15,16,17]. Recently, researchers found that approximately 23.1% of circular RNAs (circRNAs) in pancreatic ductal adenocarcinoma tissues harbored m6A modifications [18]. These modifications regulate several facets of RNA processing, including alternative splicing [19], export [20,21], stability [22,23,24], translation [25,26], and miRNA maturation [27]. m6A is the most prevalent modification in the mRNA of many eukaryotic species, including yeast [28,29,30], plants [31,32,33,34,35,36,37,38], insects [39,40], and mammals [41,42,43].

The analysis of m6A transcriptome-wide in *Arabidopsis thaliana* revealed that m6A is a highly conserved modification of mRNA in plants [32]. Using m6A-targeted antibody coupled with high-throughput sequencing, more than 70% of the m6A peaks of Can-0 and Hen-16 were consistently detected. Differential m6A methylation patterns between three organs in *Arabidopsis thaliana* were revealed by transcriptome-wide high-throughput deep m6A-seq [44]. A comparison among the three organs revealed that more than 80% of the m6A-modified transcripts were common between leaves, flowers, and roots. The study of m6A patterns in the *Arabidopsis* chloroplast and mitochondria transcriptomes shows that the selection of the m6A motifs for RNA methylation was conserved between the nucleus and organelle transcriptomes [45]. The m6A map of sea buckthorn (*Hippophae rhamnoides* Linn.) transcriptome identified 13,287 different m6A peaks between leaf under drought and in control treatment [46].

A positive correlation between m6A deposition and plant mRNA abundance was observed. In addition, severe developmental abnormalities of leaves and roots as well as altered timing of reproductive development were reported after dysregulation of m6A modification [35,36]. These findings suggest a regulatory role of m6A in plant gene expression.

## 2. Plant Viruses May Act as an Inducer to Disrupt m6A Methylation

Significant changes in overall m6A levels after virus infection indicate that the viruses have induced m6A methylation. In mammals, SARS-CoV-2 infection triggered a global increase in host m6A methylome, exhibiting altered localization and motifs of m6A methylation in mRNAs [47]. In addition, the global cellular rate of m6A methylation increased upon HIV infection [48]. In plant viruses, it has been shown that m6A level is reduced upon *Tobacco mosaic virus* (TMV) infection in *Nicotiana tabacum* [49]. Meanwhile, the m6A methylation level of the plant endogenous mRNAs can be modified upon TMV infection. In addition, *alfalfa mosaic virus* (AMV) infection increases m6A levels in *Arabidopsis* [50]. These reports indicate that plant viruses may act as an inducer to disrupt m6A methylation.

However, how do plant viruses induce m6A methylation modification? One possibility is that the virus can interact with m6A-related proteins to negatively regulate their expression level such as METTL-like, demethylase homolog and ALKB-like, methylase homolog. These proteins remained unchanged or increased, resulting in reduced m6A methylation after virus infection [49]. For example, the gene expression level of m6A related protein (the potential demethylase XM_009801708, a protein partial homology among human AlkB Homolog 5 (ALKBH5) in *Arabidopsis* might be induced by TMV infection in *Nicotiana tabacum* at 14 and 21 days [49]. Another possibility is that plant m6A methylases make their major efforts to deal with the newly invading foreign viral RNA, which results in losing sufficient energy to perform the regular methylation of the plant endogenous gene.

## 3. Plant Viral RNA Can Be the Target of m6A Methylation

Methylation of m6A was shown to be conserved in the genomic RNAs of diverse mammalian viruses [51,52,53]. In plant viruses, the presence of m6A in the genomes of two members of the *Bromoviridae* family, AMV and *cucumber mosaic virus* (CMV), has been reported [50]. In the case of AMV, viral accumulation was reduced in inoculated leaves of a m6A demethylase (atALKBH9B) mutant in *Arabidopsis*. Whereas, higher m6A levels in AMV genomic RNAs were observed in these mutant plants. This suggests that m6A modification negatively affects viral infection. They found that the CMV genome also contains m6A but that it differs from AMV. The abundance of m6A methylation in viral RNA and virus infection were modified in atALKBH9B mutant plants, which might be due to the fact that the CMV coat protein (CP) did not interact with atALKBH9B in vivo [50]. This suggests a similar feature of m6A methylation in viruses that replicate in the cytoplasm of plant cells and mammalian. However, with these data, we know little about the status of m6A methylation in the AMV and CMV genome RNA.

According to the prediction and identification of m6A methylation in viral RNA, it is mainly concentrated in the CDS region, while m6A methylation of plant endogenous gene mRNA is mostly concentrated in the 3’UTR of TMV [49]. This indicates that before the plant was successfully infected, the 5’ and 3’UTRs of the virus had been modified by the m6A methylation mechanism of the plant, resulting in unsuccessful viral replication and/or translation. In this case, m6A methylation in the CDS region has been detected in viral RNA as well, indicating that these methylations may not influence virus replication and proliferation but act as a protective mechanism for viral RNA and protect them from being degraded by RNase. To address this possibility, we may focus on the study of virus replication enzymes to check whether virus replication enzymes have evolved to adapt to recognize multiple modified RNAs. The other question that remains unclear is whether the m6A methylations affect the translation of viral RNA.

Except for the m6A methylation in the CDS region, however, we should not ignore the possibility that the intron sequence, similar to the noncoding RNA can be methylated. In addition, the intermediate strand of RNA replication can be captured by m6A methylation. The other function may be involved in processing processes such as mRNA splicing and transportation. Indeed, m6A has now been shown to regulate mRNA splicing, highlighting the value of these early viral studies in uncovering m6A functions [8,19].

On the contrary, most m6A methylation of the mRNA of the host endogenous gene displayed the typical topology found in *Arabidopsis* with one or two high peaks at the stop codon or in the 3’UTR and very low m6A signals in the 3’UTR and CDS regions, which is related to the reported m6A methylation regulating the translation process of plant genes [44]. This can be explained by the fact that the native m6A methylation is involved in regulating gene translation and meets the requirements of the regular development of plant.

m6A methylation occurs not only on RNA viruses but also on DNA viruses [54]. A novel function of m6A RNA methylation regulates the UV-induced DNA damage response, supporting a model whereby m6A RNA serves as a beacon to facilitate repair and cell survival [20]. This is also agreed with the evidence that the DNA viruses are also subjected to be modified by m6A methylation in mammals, such as Simian virus 40 (SV40) [6,7,8] and Kaposi’s sarcoma-associated herpesvirus (KSHV) [55]. The m6A methylation in plant DNA viruses remains unknown.

## 4. m6A Methylation Is One of the Defense Mechanisms against Plant Viral Infection

In mammals, RNA m6A methylation is catalyzed by a polyprotein complex composed of METTL3, METTL14, Wilms’ tumor 1-associating protein (WTAP), the human homolog of *Drosophila* Virilizer (KIAA1429) [56], and several cofactors not yet identified [57,58]. Their functions are summarized in Appendix A.

In some cases, viral infection are positively (in the case of hepatitis C virus, HCV) and negatively (in the case of Zika virus, ZIKV) regulated by knockdown of METTL3/14 and ALKBH5, or FTO (fat mass and obesity-associated protein), respectively [52,59]. Although m6A has long been known to exist in plant mRNAs, the proteins involved in m6A methylation have only recently been detected through mutant analysis, homology search, and mRNA interactome capture in *Arabidopsis thaliana* [52,59]. The review by Marlene Reichel et al., showed that the orthologues of several methylosome subunits have been identified and were shown to interact with each other. METHYLTRANSFERASE A (MTA), a METTL3 homolog that plays a critical role in plant development in *Arabidopsis*, has been identified [60,61]. The *Arabidopsis* FIP37 protein, a plant homolog of WTAP interacting with MTA both in vitro and in vivo, is essential for mediating m6A mRNA modification of key shoot meristem genes [60,62]. In the study of a protein with homology to VIRMA/KIAA1429 involved in m6A formation in mammals [56], m6A levels in the vir-1 mutant were reduced to approximately 10%, and the mutant showed aberrant formation of lateral roots and root caps as well as aberrant cotyledon development [63]. METHYLTRANSFERASE B (MTB), an orthologue of human METTL14 [63] may display enzymatic activity in *Arabidopsis* [64]. m6A levels were reduced to 50% in an inducible MTB RNAi line [63]. An additional component of the *Arabidopsis* writer complex was HAKAI, the orthologue of an E3 ubiquitin ligase. The m6A levels are reduced to 35% in hakai mutant lines without obvious phenotypes [63]. The *Arabidopsis* homolog of RBM15 is FPA, which regulates the flowering time by RNA-mediated chromatin silencing of the floral repressor FLOWERING LOCUS C (FLC) [65,66].

YT521-B homology (YTH) domain proteins are important m6A readers with established functions in animals. Plants contain 13 previously identified *Arabidopsis* YTH domain-containing proteins [31]. Two labs established the relevance of a cytoplasmic m6A-YTH regulatory module in the timing and execution of plant organogenesis [34,36]. The cytoplasmic *Arabidopsis thaliana* YTH domain proteins, EVOLUTIONARILY CONSERVED C-TERMINAL REGION2/3 (ECT2/3) are required for the correct timing of leaf formation and normal leaf morphology [36]. ECT2 has been proposed to promote m6A-dependent stability by binding the 3′untranslated regions (3′UTRs) of target genes [34] both in the nucleus and the cytoplasm.

Some researchers also demonstrated that m6A methylation is crosslinked with the stress response to heat shock in plants. Upon heat stress, YTHDF2 relocates to the nucleus where it binds to m6A sites in the 5′UTR of stress-induced transcripts, including HSP70, thereby preventing FTO from demethylation and promoting translation [67]. One study reported the relocation of *Arabidopsis* ECT2 to stress granules upon heat stress, suggesting that m6A might also play a role in plant stress response [35].

Up to now, no anti-viral activities of those identified m6A related proteins have been described in plants at present. Transcriptome-wide m6A profiling found a significant variation in the m6A modification patterns between the resistant and susceptible wheat varieties. Different m6A RNA modifications in the two varieties demonstrated regulation of gene expression and pathogen–plant interaction-related pathways [68]. The m6A demethylase activity of atALKBH9B modulates AMV infection and the m6A abundance in its genomic RNAs, indicating that plant m6A methylation might be involved in viral infection [50]. One recent study suggested that atALKBH9B is involved in viral upload to the phloem of *Arabidopsis* [69]. The *Arabidopsis* genome contains 13 homologs of *Escherichia coli* AlkB (atALKBH1-10B) [70]. ALKBH9B removes m6A from single-stranded RNA molecules in vitro. The location of atALKBH9B in cytoplasmic granules and being associated with small interfering RNA (siRNA) bodies and P bodies is suggesting that atALKBH9B m6A demethylase activity could be linked to mRNA silencing and/or mRNA decay processes [50]. Indeed, atALKBH9B affected the infectivity of AMV, but not of CMV, correlating with the ability of atALKBH9B to interact (or not) with their coat proteins. This suggests that m6A modification may represent a plant regulatory strategy to control cytoplasmic-replicating RNA viruses.

## 5. The Virus Encodes AlkB Protein to Promote Virus Infection

*Escherichia coli* AlkB proteins (members of the 2-oxoglutarate (2OG)- and Fe(II)-dependent oxygenase superfamily) are involved in DNA and RNA repair [71,72,73]. Eukaryotes usually have several proteins encoding ALKB-like genes. Nine ALKB homologs have been identified in mammals: ALKBH1-8 and FTO. Among them, ALKB5 is well studied. In HCV and ZIKV, viral titer was negatively affected when m6A modification of their genomic RNAs were regulated by the knockdown of ALKBH5 and FTO. m6A abundance in the ZIKV genome negatively affected the viral titer [52]. The production of infectious virus decreased when subjected by the depletion of FTO but not ALKBH5 [59]. The *Arabidopsis* genome contains 13 homologs (atALKBH1-10B) of *E**. coli* AlkB [70]. According to the subcellular localization assay, all these proteins display a nucleocytoplasmic localization pattern except for atALKBH1D, which localizes to the chloroplast as well, and atALKBH9B, which is exclusively cytoplasmic [70]. The function of most proteins remains unknown. It has been demonstrated that the demethylase activity of atALKBH9B modulates viral infection of AMV but not of CMV. Whereas, it appears that atALKBH10B is involved in the regulatory network of floral transition in *Arabidopsis* [33,37]. This indicates that the host RNA methyltransferase machinery may represent an additional host regulatory mechanism to counter infection by viruses.

Several plant viruses have been found to contain ALKB protein homologs or domains, suggesting a counter-defense mechanism exerted by these viruses. In 2005, plant virologists found ALKB-like domains in 22 different single-stranded RNA positive-stranded plant viruses based on protein library sequence alignments (Table 1).

Most of these viruses are members of the *Flexiviridae* family. Sequence analysis showed that the ALKB domain may be functionally conserved, related to DNA or RNA repair, and protect the viral RNA genome from methylation damage during viral infection. In 2008, experimental evidence showed that bacterial and mammalian ALKB protein-like domains are also present in the replication enzymes of many plant viruses, such as *Grapevine virus A* (GVA), *Blueberry scorch virus* (BlScV), *Blackberry virus Y* (BVY), etc. Functional study of ALKB in these plant viruses found that the ALKB domain can repair the RNA damage caused by methylation, and it plays an important role in clearing the viral genome of harmful RNA and maintaining the stability of viral RNA [74]. The genome sequence of *Black raspberry necrosis virus* (BRNV) was found to contain the ALKB domain in the gene of replicase [75]. *Burdock mottle virus* (BdMoV) RNA1 (7038 nt) contains the AlkB-like domain sequence, which is not present in proteins encoded by other known benyviruses but is found in the replication-associated proteins of viruses mainly belonging to the families *Alfaflexiviridae* and *Betaflexiviridae* [76]. The function of ALKB domains in BRNV and BdMoV have not been characterized. Interestingly, most of the viruses above encoding the ALKB domain infect woody or perennial plants, where they have to establish infections that persist for years. It has been suggested that methylation may be used in host-mediated inactivation of viral RNAs and that AlkB homologs in some plant viruses may be used to counteract such defense mechanisms [71]. Therefore, we suggest that this might be the result of long-term evolution during the interaction between the virus and the host to counter plant defense by m6A modification. In addition, scientists believe that the introduction of AlkB domain in plant viruses was probably a recent event and originated from bacteria or the evolution event of gene recombination [77]. However, no detailed study of this has been published. We propose that the case of plant viruses carrying the ALKB domain has probably come from the natural screening events during the evolution. Plant viruses recruited the sequence to their genome RNA from the host ALKB protein in order to avoid being modified by host m6A methylation, which benefits viral infection. Those plant viruses lacking the AlkB domain may have exploited different ways to confer resistance to m6A methylation.

Although the orthologs of several methylosome subunits, such as MTA, MTB, FPA, and YTH domain proteins of ETC2-4 were identified in *Arabidopsis*, there is no report to our knowledge showing that any plant viruses contains these proteins.

## 6. The Interplay between m6A Methylation and Viruses in Plant

Although accumulating experimental data contribute greatly to the understanding of the role of m6A modification in plant, the regulation mechanism during natural infection of plant viruses remains unknown. In this review, we propose a model of interplay between the m6A methylation and plant viruses (Figure 1). On the one hand, the m6A methylation system can be used as a defense mechanism for the host to resist foreign invasion upon viral invasion. The viral RNA could be methylated by METTL-like proteins, together with YTH domain proteins, affecting the stability, translation, or viral particle packaging during viral infection. Alternatively, the positions of m6A methylation in the viral RNA directly inhibit the replication conducted by virus replicase, which affects viral replication. On the other hand, viral RNA may also induce an imbalance in the m6A methylation and demethylation status of the host endogenous genes to regulate its expression. This can, in turn, indirectly influence the viral infection. However, several mysteries remain to be clarified, such as (i) how plants utilize the m6A methylation mechanism to modulate viral infection paying special attention to the relevance of proteins associated with m6A methylation, such as ALKBH5-like, METTL21A-like, and YTH domain proteins, to viral RNA stability and infection; (ii) how plant viruses act as inducers of responses that disrupt the reversible balance of the m6A methylation system, resulting in m6A methylation changes in host genes during plant viral infection.

## 7. Conclusions and Perspectives

Given the reason for few reports about m6A methylation in plant viruses, the study of m6A modification in plant viruses not only reveals a new layer of epigenetic regulation in viruses but also provides potential molecular mechanisms for viral infection, plant immune response, and the designing of anti-viral resistance. Here, we point out several theoretical research topics that remain to be clarified in the future. From the virus part, one should make a great effort to address the factors participating in the interplay between m6A methylation and plant virus infection, not only in plant RNA viruses but also in DNA viruses. In particular, the CP protein is the first protein that encounters the plant immune response after virus invasion. The process of CP recognition with the m6A methylation is an interesting question. The methylation characteristics of the secondary structure of viral RNAs and influence on viral infection remains unclear at present. According to the fact of several plant viruses containing ALKB domain, we open the possibility of ALKB domains being involved in the m6A methylation machinery. Viruses containing ALKB domains should be used to test this hypothesis. From the plant side, does m6A methylation affect viral replication or translation? Does M6A methylation promote or inhibit viral RNA degradation? Does m6A methylation cross-link with other immune pathways to confer anti-viral resistance?

Regarding the potential application of m6A methylation in plant virus control strategies, the following aspects can be considered: (i) with the improved m6A-seq technology, early detection of plant viral infection can be done with plant tissues or plant protoplasts for m6A modification level analysis. The m6A modification map of certain specific transcripts or transcript loci can be used as biomarkers for early viral diagnoses, classifications, outcome predictions, and risk grading; (ii) identification of m6A-modified targets in viruses may help us to directly design the m6A-directed anti-viral strategies to confer viral resistance to plant; (iii) we should also focus on the m6A modification on ncRNAs in plant viruses. To study how m6A modification affects ncRNAs production, function, processing, which indirectly regulates viral infection? This will greatly broaden our understanding of the unknown functions of m6A-modified ncRNA in plants; (iv) selective small molecule inhibitors should be an extremely attractive disease control strategy in plant viruses. Indeed, some pioneering proof-of-concept studies have shown that targeting dysregulated m6A methylase through small molecule inhibitors has the potential to treat cancer. Some small pharmaceutical or biotechnology companies such as STORM Therapeutics, Accent Therapeutics (co-founded by He Chuan and Howard Chang Zhang Yuanhao), Gotham Therapeutics and Genovel Biotech Corp have begun development, targeting m6A methylases such as METL3, METL14, and FTO; (v) we also suggest that new strategies in controlling plant viruses could be designed through the competitive antagonists of m6A-associated proteins or engineered molecules to confer anti-viral resistance in the applications in the future.

## Figures and Tables

**Figure 1 biology-11-00271-f001:**
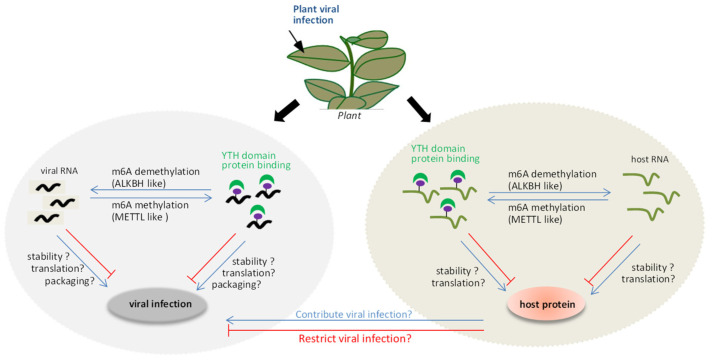
Schematic overview of virus-induced regulation of m6A methylation. When viral RNAs invade into plant cells, the m6A RNA methylase, demethylase and YTH domain proteins will modify the viral RNA. This might directly affect viral RNA stability, translation, and viral particle packaging. Alternatively, viral infection indirectly influences the expression level of host antiviral proteins, which could contribute or restrict viral infection.

**Table 1 biology-11-00271-t001:** Plant viruses involved in m6A methylation or contained ALKB domains.

Virus.	M6A-Related Proteins	Summary of Knowledge	References
*Aalfalfa mosaic virus* (AMV)	ALKBH9B (At2g17970)	The demethylation activity of atALKBH9B affected the infectivity of AMV by interacting with CP of AMV. Suppression of atALKBH9B increased the relative abundance of m6A in the AMVgenome, impairing the systemic invasion of the plant.	Martinez-Perez et al., 2017
*Cucumber mosaic virus* (CMV)	ALKBH9B (At2g17970)	atALKBH9B does not have any effect on CMV infection. atALKBH9B does not interact with CP of CMV.	Martinez-Perez et al., 2017
*Tobacco mosaic virus* (TMV)	The potential demethylase XM_009801708 in *Nicotiana tabacum.*	The overall level of m6A decreases after (TMV) infection in *Nicotiana tabacum*. The expression level of XM_009801708 is increased upon TMV infection.	Zhurui et al., 2018
*Grapevine virus A* (GVA)	Containing ALKB domain in viral genome	Maintaining the integrity of the viral RNA genome through removal of deleterious RNA damage.	Van den Born et al., 2008
*Blueberry scorch virus* (BlScV)	Containing ALKB domain in viral genome	Maintaining the integrity of the viral RNA genome through removal of deleterious RNA damage.	Van den Born et al., 2008
*Blackberry virus**Y* (BVY)	Containing ALKB domain in viral genome	Maintaining the integrity of the viral RNA genome through removal of deleterious RNA damage.	Van den Born et al., 2008
*Little cherry virus* (LChV-2)	Containing ALKB domain in viral genome	Not tested.	Van den Born et al., 2008
*Citric leave blotch virus* (CLBV)	Containing ALKB domain in viral genome	Not tested.	Van den Born et al., 2008
Chrysanthemum virus B (CVB)	Containing ALKB domain in viral genome	Not tested.	Van den Born et al., 2008
Lily symptomless virus (LSV)	Containing ALKB domain in viral genome	Not tested.	Van den Born et al., 2008
*Apple stem pitting* virus (ASPV)	Containing ALKB domain in viral genome	Not tested.	Van den Born et al., 2008
*Garlic latent virus* (GLV)	Containing ALKB domain in viral genome	Not tested.	Van den Born et al., 2008
*Zygocactus virus**X* (ZVX)	Containing ALKB domain in viral genome	Not tested.	Van den Born et al., 2008
*Burdock mottle virus* (BdMoV)	Containing ALKB domain in viral genome	Not tested.	Kondo et al., 2013
*Black raspberry necrosis virus* (BRNV)	Containing ALKB domain in viral genome	Not tested.	McGavin et al., 2010

## Data Availability

The study did not report any data.

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
