# Peer review of "The Reversible Methylation of m6A Is Involved in Plant Virus Infection"

_biology, 2022, doi:10.3390/biology11020271_

Round 1

Reviewer 1 Report

Yue et al. submitted a Review article titled, "The reversible methylation of m6A is involved in plant virus 2 infection", for publication in MDPI Biology. 

The manuscript is well written and qualifies for publication. However, please make these minor corrections.

  1. Italicize all names of viruses everywhere in the ms.
  2. Line 261 - Change ALKB to AlkB
  3. Line 269 - Modify ZIKA to ZIKV.
  4. Line 274 - Italicize organism names here and everywhere in the ms. (Escherichia coli)
  5. Add detailed description of Fig. 1. below it.

Author Response

Point to point response for reviewers

According to the reviewer’s comments, we have made major revisions in the manuscript. Some paragraph was deleted. Thus, the numbers of lines are not corresponding to previous submitted version of manuscript. We have marked the requirements by green color. The point to point responses are listed as below.

Reviewer 1:

  1. Italicize all names of viruses everywhere in the ms.

--Reply by author: thank you for reviewer’s comment. We have italicized all names of viruses everywhere and marked in green color in the manuscript.

  1. Line 261 - Change ALKB to AlkB.

--Reply by author: thank you for reviewer’s comment. In line 216, we have changed the ALKB to AlkB and marked in green color in the manuscript. 

  1. Line 269 - Modify ZIKA to ZIKV.

--Reply by author: thank you for reviewer’s comment. In line 269, we have modified the ZIKA to ZIKV and marked in green color in the manuscript.

  1. Line 274 - Italicize organism names here and everywhere in the ms. (Escherichia coli).

--Reply by author: thank you for reviewer’s comment. We have italicized organism names of Escherichia coli in line 274 and everywhere and marked in green color in the manuscript.

  1. Add detailed description of Fig. 1. below it.

--Reply by author: thank you for reviewer’s comment. We have added description of Figure. 1 as “When viral RNAs invade into plant cell, the m6A RNA methylase, demethylase and YTH domain proteins are going to modify the viral RNA. This might directly affect viral RNA stability, translation and viral particle packaging. Or viral infection is indirectly influencing the expression level of host antiviral proteins, which could contribute or restrict viral infection”. below it.

Reviewer 2 Report

Dear authors, your manuscript on RNA methylation give a good view on the implication of RNA methylation in the interaction between hosts and viruses. However, it will be great if you can add a complementary table with the name or function of the protein cited in the text (METTL, WTAP, KIAAALKBH, FTO….).  There is also few changes to be made:

Line 126 and 130 alkbh9b should be written in capital

Line 138 the virus was successfully infected should be change to ‘the plant was ….

Line 158-159. Coding regions and CDS regions have the same meaning. Cancel one

Author Response

Point to point response for reviewers

According to the reviewer’s comments, we have made major revisions in the manuscript. Some paragraph was deleted. Thus, the numbers of lines are not corresponding to previous submitted version of manuscript. We have marked the requirements by green color. The point to point responses are listed as below.

Reviewer 2:

  1. Line 126 and 130 alkbh9b should be written in capital.

--Reply by author: thank you for reviewer’s comment. In line 126 and 130, alkbh9b has been revised in capital and marked in green color in the manuscript.

  1. Line 138 the virus was successfully infected should be change to ‘the plant was ….

--Reply by author: thank you for reviewer’s comment. In line 138, we have modified “the virus was successfully infected” to “the plant was successfully infected” and marked in green color in the manuscript.

  1. Line 158-159. Coding regions and CDS regions have the same meaning. Cancel one.

--Reply by author: thank you for reviewer’s comment. line 158-159, we have deleted coding regions and marked in green color in the manuscript.

  1. However, it will be great if you can add a complementary table with the name or function of the protein cited in the text (METTL, WTAP, KIAA, ALKBH, FTO….).

-- Reply by author: thank you for reviewer’s comment. We have created a complementary table with the name or function of the protein cited in the text (METTL, WTAP, KIAA, ALKBH, FTO….) in supplementary Table 1.

Reviewer 3 Report

In this manuscript, authors describe and discuss current and past research on a specific modification (m6A) occurring at mRNA of many eukaryotic species and relate it with a possible biological function in plant virus infection. The m6A modification is only recently investigated in plants, and its role during virus infection, as well as other responses to environmental stimulus, are of interest for the plant science community.

However, I found this review extremely unbalanced. From one side there is excess and redundancy of description of the m6A role in mammal models. Although the research on m6A in mammal virus infection is better investigated, the references to mammal models are excessive and often not relevant (e.g.  potential therapeutic application on human diseases is irrelevant for plants). There are often changes of the discussion from plant to mammal models, making unclear what is actually known in plants (examples in chapter 3 and 4). It would be beneficial to group in the same chapter a general description of m6A regulation (as this is mostly known from mammals) and then try to focus on plant m6A and plant viruses in the rest of the manuscript.

On other hand, the details given on m6A regulation in plants and its potential involvement in viral infection are confused and at several points unclear. Some critical definitions are not given (e.g. METTL and ALKB are not properly introduced), and authors attempt speculations (at end of chapter 5 and in chapter 6) that are not supported by their discussion (e.g. based on which data authors suggest that the anti-defence function of ALkB homologs in plant viruses is the result of long-term evolution? What authors means with natural screening events?). There are also cases where information given is not supported by enough evidence. For examples, in chapter 2 the second paragraph contains claims not supported by any reference. Or in chapter 3 the m6A role during DNA virus infection in plants is cited without evidence.

Finally, another problem of the manuscript is the number of typos and occurrences of unclear syntax constructs (especially starting from chapter 2). Verbs are often not correctly conjugated, sentences are sometime too complex or inconclusive, or contain colloquial forms and technical imprecisions. Just to make examples, name of virus or plant species are not written in italic and same is for names of mutant (e.g. atalkbh9b), and Arabidopsis findings are automatically generalised to all plants. Also, the reference list is not organised in alphabetical order. Collectively, these mistakes, together with the confusing structure of the review, make challenging for the reader to follow authors reasoning, and I am afraid in the current form the manuscript is not helping much to build a picture of the function of m6A during plant viral infection.

Author Response

Point to point response for reviewers

According to the reviewer’s comments, we have made major revisions in the manuscript. Some paragraph was deleted. Thus, the numbers of lines are not corresponding to previous submitted version of manuscript. The point to point responses are listed as below.

Reviewer 3:

  1. However, I found this review extremely unbalanced. From one side there is excess and redundancy of description of the m6A role in mammal models. Although the research on m6A in mammal virus infection is better investigated, the references to mammal models are excessive and often not relevant (e.g. potential therapeutic application on human diseases is irrelevant for plants). There are often changes of the discussion from plant to mammal models, making unclear what is actually known in plants (examples in chapter 3 and 4). It would be beneficial to group in the same chapter a general description of m6A regulation (as this is mostly known from mammals) and then try to focus on plant m6A and plant viruses in the rest of the manuscript.

--Reply by author: thank you for reviewer’s comments. You are right that research on m6A in mammal virus infection is better investigated. According to reviewer’s suggestion, we are revised each chapter and focus on m6A methylation in plant viruses. Those information in mammal virus or in plants were only used as introduction to plant viruses.

  1. On other hand, the details given on m6A regulation in plants and its potential involvement in viral infection are confused and at several points unclear. Some critical definitions are not given (e.g. METTL and ALKB are not properly introduced), and authors attempt speculations (at end of chapter 5 and in chapter 6) that are not supported by their discussion (e.g. based on which data authors suggest that the anti-defence function of ALkB homologs in plant viruses is the result of long-term evolution? What authors means with natural screening events?). There are also cases where information given is not supported by enough evidence. For examples, in chapter 2 the second paragraph contains claims not supported by any reference. Or in chapter 3 the m6A role during DNA virus infection in plants is cited without evidence.

--Reply by author: thank you for reviewer’s comments.

  • We have tried to clearly describe the details given on m6A regulation in plants and its potential involvement in viral infection.
  • Some critical definitions , such asMETTL and ALKBare described in the manuscript.
  • Regarding with reviewer’s comments “the speculations (at end of chapter 5 and in chapter 6) that are not supported by their discussion (e.g. based on which data authors suggest that the anti-defense function of ALkB homologs in plant viruses is the result of long-term evolution? What authors means with natural screening events?)”, we have modified the title of Chapter 5 to “5. The virus encodes the AlkB protein to promote virus infection”. Below, we described those viruses containing AlkB protein in viral genome. We use the reports of AlkB in E.coli and Arabidopsis as the introduction to introduce the AlkB domains in plant viruses. Especially, “several plant viruses have been found to contain ALKB protein homologs or domains, suggesting a counter-defense mechanism exerted by these viruses”. Next, according to the evidences of AlkB protein in plant viruses, “we suggest that this might be the result of long-term evolution during the interaction between the virus and the host to counter plant defense by m6A modification. In addition, scientists believed that the introduction of AlkB domain in plant virus was probably a recent event and originated from bacteria or evolution event of gene recombination. However, no detailed study of this has been published. We propose that the case of plant viruses carrying ALKB domain is probably come from the natural screening event during the evolution. Plant viruses recruited the sequence to their genome RNA from the host ALKB protein in order to avoid been modified by host m6A methylation, which benefit viral infection. Those plant viruses lacking the AlkB domain may have exploited different ways to confer resistance to m6A methylation”.These are speculation and discussion points raised by the author. No reports were found yet. However, these points are really interesting and deserved to be addressedin the future.
  • in chapter 2 , we have add the reference[49]in the corresponding sentences. The sentence of “Another explanation is that after the virus invaded the host plant, the plant methylase make their major efforts to deal with the newly invading foreign viral RNA without sufficient energy to perform the regular methylation of the plant endogenous gene” is speculated by author.
  • in chapter 3, no report was found about the m6A role during DNA virus infection in plants. This is also an interesting point for scientists who are working with the plant DNA viruses.  

  1. Finally, another problem of the manuscript is the number of typos and occurrences of unclear syntax constructs (especially starting from chapter 2). Verbs are often not correctly conjugated, sentences are sometime too complex or inconclusive, or contain colloquial forms and technical imprecisions. Just to make examples, name of virus or plant species are not written in italic and same is for names of mutant (e.g. atalkbh9b), and Arabidopsis findings are automatically generalised to all plants. Also, the reference list is not organised in alphabetical order. Collectively, these mistakes, together with the confusing structure of the review, make challenging for the reader to follow authors reasoning, and I am afraid in the current form the manuscript is not helping much to build a picture of the function of m6A during plant viral infection.

--Reply by author: thank you for reviewer’s comments.

  • We have revised the typos and those verbs which are often not correctly conjugated. We have simplifiedthe sentences and those problems of the colloquial forms and technical imprecisions.
  • We have revised the problems, such as those namesof virus or plant species are not written in italic.
  • Regarding with “The names of mutant (e.g. atalkbh9b)”, we have revised to “atALKBH9B mutant plants. atALKBH9Bis also described.
  • Also, we have corrected the reference list.

Reviewer 4 Report

Review of “The reversible methylation of m6A is involved in plant virus infection.” This manuscript is in need of some serious attention. Unfortunately, the crux of the paper is lost in the myriad of references that are not explained nor connected to the central subject by the authors. Many times, a paragraph begins discussing one thing and abruptly, with no transition, jumps to another. Somehow the reader is supposed to connect the dots that the authors were intending to convey. Here are a few more specific issues:

The document as a whole, needs to be edited by one person who is fluent in English grammar and another who is not associated to the field of study. The grammar is quite poor, with innumerable instances of misused verb tenses, erroneously chosen words and missing articles (such as: the, and, an..). The sentence structures need to be revised so that it is more understandable and then it may be able to connect those seemingly unrelated topics mentioned earlier. Additionally, there are instances where the punctuation is inaccurate, including double periods at the end of sentences and spaces before periods at the end of sentences.

The second person needed is due to the fact that things that become second nature in the authors’ field may become overlooked and assumed to be known by all in the scientific community and therefore omitted from the text inadvertently.

All abbreviations within the text need to be spelled out with the initial use. There are several instances where this is not the case (circRNAs, atalkbh9b, siRNA, etc.) and need to be addressed. There are also instances where a term is all uppercase in one sentence and all lowercase in the next sentence. Please ensure continuity in the text.

Since this is a Review paper, considering a multitude of diverse manuscripts from various scientific fields, there should be no reference to “unpublished data” since this will be impossible for anyone reading it once published to obtain any context to that information. If you feel the data is of utmost importance, it may be best to publish the unpublished data first before publishing a review so that you can confidently include it to your narrative.

On page 6, line 288, Table 1 is listed as a reference but there is no table in the manuscript. If the authors are referring to the supplementary material, this needs to be specified so that readers understand. Also, with this same sentence, why would a journal reference be following the Table 1 reference? Is this table from a different source?

On page 7, line 326, is a reference to Figure 1, but the figure is not presented near this paragraph. It is located 3.5 paragraphs later on page 8, line 365-385. This ought to be visible near the text. In addition, the figure legend on Figure 1 is minimal at best and requires extrapolation to explain the context of what the authors are trying to convey.

Author Response

Point to point response for reviewers

According to the reviewer’s comments, we have made major revisions in the manuscript. Some paragraph was deleted. Thus, the numbers of lines are not corresponding to previous submitted version of manuscript. The point to point responses are listed as below.

Reviewer 4:

  1. The document as a whole, needs to be edited by one person who is fluent in English grammar and another who is not associated to the field of study. The grammar is quite poor, with innumerable instances of misused verb tenses, erroneously chosen words and missing articles (such as: the, and, an..). The sentence structures need to be revised so that it is more understandable and then it may be able to connect those seemingly unrelated topics mentioned earlier. Additionally, there are instances where the punctuation is inaccurate, including double periods at the end of sentences and spaces before periods at the end of sentences.

--Reply by author: thank you for reviewer’s comments.

  • The document as a wholehas been edited by Airong Liwho is fluent in English grammar and another who is not associated to the field of study. Her research is mainly in parasitic plants.
  • We have revised the sentences in the manuscriptand try to explain clearly the ideas.
  • The problems about the inaccurate punctuationare also corrected.

  1. The second person needed is due to the fact that things that become second nature in the authors’ field may become overlooked and assumed to be known by all in the scientific community and therefore omitted from the text inadvertently.

--Reply by author: thank you for reviewer’s comments. It’s a nice suggestion that we should ask for the second person who have difference knowledge could help us to complete the review. Airong Li who is familiar with the parasitic plants has revised the manuscript.

  1. All abbreviations within the text need to be spelled out with the initial use. There are several instances where this is not the case (circRNAs, atalkbh9b, siRNA, etc.) and need to be addressed. There are also instances where a term is all uppercase in one sentence and all lowercase in the next sentence. Please ensure continuity in the text.

--Reply by author: thank you for reviewer’s comments. All abbreviations within the text have been spelled out with the initial use, for example, circRNAs, atalkbh9b, siRNA, etc. We try to revise the instance where a term is all uppercase in one sentence and all lowercase in the next sentence. Taken AlkB as an example, when people talking about in AlkB family, it was written in “AlkB”. However, when we described the gene’s name, it was written in “ALKB” (Duan HC, Wang Y, Jia G. Dynamic and reversible RNA N6 -methyladenosine methylation. Wiley Interdiscip Rev RNA. 2019;10(1):e1507. doi:10.1002/wrna.1507).Therefore, we would like follow this rule in the manuscript.

  1. Since this is a Review paper, considering a multitude of diverse manuscripts from various scientific fields, there should be no reference to “unpublished data” since this will be impossible for anyone reading it once published to obtain any context to that information. If you feel the data is of utmost importance, it may be best to publish the unpublished data first before publishing a review so that you can confidently include it to your narrative.

--Reply by author: thank you for reviewer’s comments. The “unpublished data” mentioned in the manuscript are under submission and not published yet. As suggested by reviewer, we have deleted the “unpublished data”.

  1. On page 6, line 288, Table 1 is listed as a reference but there is no table in the manuscript. If the authors are referring to the supplementary material, this needs to be specified so that readers understand. Also, with this same sentence, why would a journal reference be following the Table 1 reference? Is this table from a different source?

--Reply by author: thank you for reviewer’s comments. We have added the Table 1 in the manuscript. We created the Table 1 to summarize the reported knowledge about plant viruses involved in m6A methylation or contained ALKB domains. We thought this will be clearly and useful for readers to obtain the information and source of references.

  1. On page 7, line 326, is a reference to Figure 1, but the figure is not presented near this paragraph. It is located 3.5 paragraphs later on page 8, line 365-385. This ought to be visible near the text. In addition, the figure legend on Figure 1 is minimal at best and requires extrapolation to explain the context of what the authors are trying to convey.

--Reply by author: thank you for reviewer’s comments. We have created a chapter to describe the ideas of figure 1. We have added description of Figure 1 as “When viral RNAs invade into plant cell, the m6A RNA methylase, demethylase and YTH domain proteins are going to modify the viral RNA. This might directly affect viral RNA stability, translation and viral particle packaging or indirectly influencing the expression level of host antiviral proteins”.

Round 2

Reviewer 4 Report

2nd Review of “The reversible methylation of m6A is involved in plant virus infection”

It does appear that the authors have put effort into modifying their manuscript. Abbreviations have been spelled out, unpublished data has been removed and Figure 1 has been updated with a descriptive legend. There has been several grammar and structure edits, but some were overlooked. I point these out merely because it is best to have a manuscript at 100% when publishing so that readers will not stumble or need to re-read sentences. I’ve listed the lines that nevertheless need particular corrections, including articles, tenses and word usages. Some need restructuring as well. The addition of Table 1 was quite useful and appreciated. I realize it was undoubtedly just a formatting matter, however, the header for the table was situated at the top of page 5 instead of on the table itself on page 6. I understand that programs occasionally don’t do what we want them to do, but just in case, you may want to check it. Overall, it is considerably more understandable and clearer. The flow is improved and with the changes, it is easier to comprehend. Also, the concluding paragraph in the conclusions and perspectives section is a brilliant improvement. It provides a great deal more understanding into the future of m6A investigations and prospective applications. A worthy conclusion.

Lines from most recent copy that need grammatical and or structure edits:

20, 49, 78, 80, 103, 117, 119-120, 126, 129, 144, 145, 149, 158, 165, 196, 206, 207, 215, 246, 265, 276, 277, 287, 318, 322, 338, 339

Author Response

Dear Editor,

We would like to submit a revised manuscript entitled "The reversible methylation of m6A is involved in plant virus infection". This manuscript has been appropriately revised using MDPI's "English Editing" service as you suggested.

Here is the reply what you mentioned in email on 17th January 2022:

- Please use the version of your manuscript found at the above link for your revisions.

---Reply by author: In this manuscript, we modified using the version found at this link: https://susy.mdpi.com/user/manuscripts/resubmit/545ffb2580db572ca86c3aa4c2cf58c2

- Any revisions made to the manuscript should be marked up using the “Track Changes” function if you are using MS Word/LaTeX, such that changes can be easily viewed by the editors and reviewers.

---Reply by author: We use the “Track Changes” function in the revised manuscript.

Please provide a short cover letter detailing your changes for the editors’ and referees’ approval.

---Reply by author: A short cover letter is provided including the sentences that need grammatical or structure edits.

Here is the reply what you mentioned in email on 20th January 2022:

Should the references style in Table 1 been changed into ACS style?

--Reply by author: The references involved in Table 1 are listed in ACS style in the references at the end of the article, so only abbreviations are used here. We thought it will be easy to track in references.

Comments and Suggestions for Authors:

2nd Review of “The reversible methylation of m6A is involved in plant virus infection”

It does appear that the authors have put effort into modifying their manuscript. Abbreviations have been spelled out, unpublished data has been removed and Figure 1 has been updated with a descriptive legend. There has been several grammar and structure edits, but some were overlooked. I point these out merely because it is best to have a manuscript at 100% when publishing so that readers will not stumble or need to re-read sentences. I’ve listed the lines that nevertheless need particular corrections, including articles, tenses and word usages. Some need restructuring as well. The addition of Table 1 was quite useful and appreciated. I realize it was undoubtedly just a formatting matter, however, the header for the table was situated at the top of page 5 instead of on the table itself on page 6. I understand that programs occasionally don’t do what we want them to do, but just in case, you may want to check it. Overall, it is considerably more understandable and clearer. The flow is improved and with the changes, it is easier to comprehend. Also, the concluding paragraph in the conclusions and perspectives section is a brilliant improvement. It provides a great deal more understanding into the future of m6A investigations and prospective applications. A worthy conclusion.

Lines from most recent copy that need grammatical and or structure edits: 20, 49, 78, 80, 103, 117, 119-120, 126, 129, 144, 145, 149, 158, 165, 196, 206, 207, 215, 246, 265, 276, 277, 287, 318, 322, 338, 339

--- Reply by author: Thanks for Reviewer4’s comments. Here are the point by point revisions.

Line 20: “viral infection in human system” has been modified to “viral infection in the human system”.

Line 49: “This led the virologists” have been modified to “This led virologists”.

Line 78: “the m6A modified transcripts” have been modified to “the m6A-modified transcripts”.

Line 80: “shows that selection of the” have been modified to “shows that the selection of the”.

Line 117: “This is suggesting that m6A” have been modified to “This suggests that m6A”.

Line 119: “but differs from” have been modified to “but that it differs from”.

Line 120: “due to the the CMV” have been modified to “due to the fact that the CMV”.

Line 127: “concentrated in 3'UTR” has been modified to “concentrated in the 3'UTR”.

Line 129: “mechanism of plant, resulting unsuccessful” have been modified to “mechanism of the plant, resulting in unsuccessful”.

Line 144: “most of the m6A methylation on the mRNA ” have been modified to “most m6A methylation of the mRNA ”.

Line 149: “and meet the requirement of” have been modified to “and meets the requirements of”.

Line 165: “or fat mass and obesity-associated (FTO)” have been modified to “or FTO (fat mass and obesity-associated protein)”.

Line 196: “transcripts including HSP70” have been modified to “transcripts, including HSP70”.

Line 207: “might been involved in” have been modified to “might be involved in”.

        “recent research suggests that” have been modified to “recent research suggested that”.

Line 215: “of AMV but not of CMV” have been modified to “of AMV, but not of CMV”.

Line 265: “AlkB homologues” have been modified to “AlkB homologs”.

Line 276: “Although the orthologues” have been modified to “Although the orthologs”.

Line 277: “there is no report showing that any plant viruses contains these proteins to our knowledge ” have been modified to “there is no report to our knowledge showing that any plant viruses contains these proteins”.

Line 287: “affect the stability”have been modified to “affecting the stability”.

        “with the YTH domain” have been modified to “with YTH domain”.

Line 318: “indirectly influence the” have been modified to “indirectly influences the”.

Line 322: “plant viruses not only reveals” have been modified to “plant viruses, not only reveals”.

Line 338: “strategy” has been modified to “strategies”.

Line 339: “can be considered.” have been modified to “can be considered:”

Thank you very much for your time.

Sincerely yours,

Mingmin Zhao